# Large-Scale Detection of the Tableland Areas and Erosion-Vulnerable Hotspots on the Chinese Loess Plateau

**Kai Liu** [1], **Jiaming Na** [2], **Chenyu Fan** [1], **Ying Huang** [3], **Hu Ding** [4], **Zhe Wang** [3], **Guoan Tang** [3] **and Chunqiao Song** [1,*]

1    Key Laboratory of Watershed Geographic Sciences, Nanjing Institute of Geography and Limnology, Chinese Academy of Sciences, Nanjing 210008, China; kliu@niglas.ac.cn (K.L.); fanchenyu21@mails.ucas.ac.cn (C.F.)
2    College of Civil Engineering, Nanjing Forestry University, Nanjing 210037, China; jiaming.na@njfu.edu.cn
3    School of Geography, Nanjing Normal University, Nanjing 210023, China; 01180535@njnu.edu.cn (Y.H.); 10180306@njnu.edu.cn (Z.W.); tangguoan@njnu.edu.cn (G.T.)
4    School of Geography, South China Normal University, Guangzhou 510631, China; hu_ding@m.scnu.edu.cn
*    Correspondence: cqsong@niglas.ac.cn

**Abstract:** Tableland areas, featured by flat and broad landforms, provide precious land resources for agricultural production and human settlements over the Chinese Loess Plateau (CLP). However, severe gully erosion triggered by extreme rainfall and intense human activities makes tableland areas shrink continuously. Preventing the loss of tableland areas is of real urgency, in which generating its accurate distribution map is the critical prerequisite. However, a plateau-scale inventory of tableland areas is still lacking across the Loess Plateau. This study proposed a large-scale approach for tableland area mapping. The Sentinel-2 imagery was used for the initial delineation based on object-based image analysis and random forest model. Subsequently, the drainage networks extracted from AW3D30 DEM were applied for correcting commission and omission errors based on the law that rivers and streams rarely appear on the tableland areas. The automatic mapping approach performs well, with the overall accuracies over 90% in all four investigated subregions. After the strict quality control by manual inspection, a high-quality inventory of tableland areas at 10 m resolution was generated, demonstrating that the tableland areas occupied 9507.31 km$^2$ across the CLP. Cultivated land is the dominant land-use type on the tableland areas, yet multi-temporal observations indicated that it has decreased by approximately 500 km$^2$ during the year of 2000 to 2020. In contrast, forest and artificial surfaces increased by 57.53% and 73.10%, respectively. Additionally, we detected 455 vulnerable hotspots of the tableland with a width of less than 300 m. Particular attention should be paid to these areas to prevent the potential split of a large tableland, accompanied by damage on roads and buildings. This plateau-scale tableland inventory and erosion-vulnerable hotspots are expected to support the environmental protection policymaking for sustainable development in the CLP region severely threatened by soil erosion and land degradation.

**Keywords:** loess tableland; landform mapping; gully erosion; land degradation; Chinese Loess Plateau; remote sensing

## 1. Introduction

The Chinese Loess Plateau is the most extensive soil deposit region globally, with an area of 640,000 km$^2$ [1–3]. Numerous gullies developed on the Loess Plateau caused severe soil erosion, land degradation, and ecosystem instability [4,5]. According to the different morphologies of gully-affected areas, the landforms between valleys can be divided into three major types: loess tablelands, loess ridges, and loess hills [6]. Among them, loess tablelands are featured by the flat high plains with very thick loess layers, providing precious land resources for the human activity of settlement, farming, and industrial production (Figure 1) [7]. Long-term observations suggest that continuous gully erosion

triggered by extreme weather conditions and unreasonable human activities encroached on tableland areas [8,9]. The shrinkage of loess tablelands has posed a direct threat to agronomic productivity and the ecological environment. Consequently, preventing or alleviating tableland shrinkage and degradation is an urgent action to protect land resources as well as support sustainable development on the Loess Plateau [10,11].

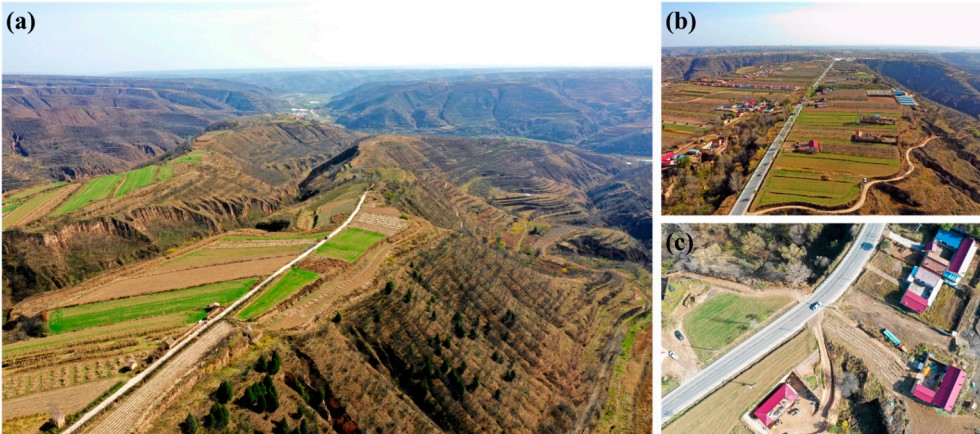

**Figure 1.** Photos of loess tableland taken by UAV-mounted cameras. (**a**) Tableland areas are featured by flat terrains surrounded by deep-cut gullies; (**b**) tableland areas provide a precise land resource for agricultural production and living settlements; (**c**) gully development may cause the shrinkage of tableland areas and even cut the large tableland areas into several disconnected parts.

Many previous efforts have been paid to investigate the status, ongoing challenges, and potential drivers of the shrinkage of tableland areas at different spatial scales. Because of the high accessibility to the long-term and multi-variable datasets, the catchment–scale studies dominated the published literature on this issue, including gully erosion and landslide monitoring [7], sediment load estimation [12], and simulating the hydrology response to climate change [13], which enable us to better understand the physical processes and driving mechanisms of tableland shrinkage. Meanwhile, a few studies also emphasize spatial heterogeneity of tablelands by implementing board-scale studies, including landform evolution modeling [2], soil erosion assessment and control [8,14], and probing the optimal mode of ecological restoration [5,11].

Generating an accurate distribution map of tableland areas is critical for the above studies, ranging from catchment scale to plateau scale. Although the field investigation or the manual interoperation based on the topographic sheet and aerial map can provide accurate distribution maps, such heavy and time-consuming work is challenging to apply at a large scale [15]. With the increasing access and sources of satellite-based earth observations (EO), remote sensing imagery and digital elevation models (DEMs) were widely applied in extracting specific landforms [16–19]. The automatic approaches for tableland area mapping mainly contain two different strategies: (1) the topographic method, which relies on the solid relationship between the topographic attribute and gully distribution [6,20]; and (2) the remote sensing image processing method, which considers the spectral similarities [21]. Due to the advantages of enriching data sources and spectrum information, the remote sensing method is more creditable, especially using the framework integrating objected-based image analysis and machine learning algorithm [22–25]. However, the application of the existing method in large-scale tableland mapping is still restricted by poor accuracy [26]. On the one hand, the commonly used methods were mainly developed at catchment scales in which the strong spatial heterogeneity of the tableland landform has not been considered. For example, the tableland areas near the Wei River remain very complete due to the slight gully erosion. The predicted model trained in this region is difficult to apply to the northern Shaanxi or western Shanxi, where tableland areas have been eroded into fragmented landforms. On the other hand, the widespread agriculture ter-

races make the extraction of tableland areas complex [27]. The simple removal of detected gully-affected areas does not match well with tableland areas [28]. Such disadvantages lead to a lack of plateau-scale inventory of tableland areas, and its spatial distribution pattern remains unclear.

This study aims to generate a high-quality inventory of tableland areas across the Chinese Loess Plateau and reveal its spatial patterns in terms of total area, density, and morphological characteristics towards the protection and ecological restoration of tableland areas. A novel approach was designed for large-scale tableland area mapping using a two-step classification. The first step of the proposed method is similar to the existing remote sensing method in which object-based image analysis and random forest algorithms were used. After achieving the initial classification results, we further used the drainage networks derived from the DEMs to correct the classification results. The topographic correction strategy is based on the truth that rivers or streams are unlikely to occur in tableland areas due to the flat terrain. Based on the automatic mapping results, rigorous manual inspection was conducted to generate high-quality distribution maps of tableland areas which have been freely available to the public. In addition, we also provided a comprehensive list of erosion-vulnerable regions which may be split by gully development. The final products will provide a valuable reference to prevent the loss of tableland areas and further promote the ecological restoration and economic development of the Loess Plateau.

## 2. Study Area and Materials

### 2.1. Study Area

The Loess Plateau of China is located between $100°52'–114°33'$E and $33°41'–41°16'$N and is the largest loess deposit region in the world, with an area of approximately 640,000 km$^2$ [3]. As a typical loess landform, the loess tableland is mainly distributed in the southern part of the Loess Plateau. In this study, we determined the study area by referring to the existing regionalization map of the Loess Plateau (Supplementary Figure S1), which delineates the regions that belong to loess tableland. To provide a comprehensive distribution map of loess tableland areas, a 20 km buffer and the watershed delineated using a 10 km$^2$ threshold value were jointly used to determine the final study area (Figure 2). Hence, the study area is litter larger than the loess tableland region derived from the existing landform atlas. It should be noted that the geomorphological features of the tableland areas within the study area exhibit strong spatial heterogeneity due to the different gully development stages. For example, the gully density in the northern part of the study area is much higher than the southern areas. In this study, we divided the whole study area into four subregions based on the landform similarity, including eastern Gansu (EGS), northern Wei River (NWR), northern Shaanxi (NSX), and western Shanxi (WSX). The predicted model was constructed and applied within four subregions independently so that uncertainties caused by the geomorphic difference can be largely reduced. In addition, the Sentinel images covering the study area are too large to run the segmentation algorithm. Zonal processing is an effective way to handle the large data volume.

### 2.2. Materials

The adopted datasets for tableland area mapping were the Sentinel-2 imagery and AW3D30 DEM. In addition, land-use data (GlobeLand30) were also employed for land-use change analysis.

#### 2.2.1. Sentinel-2 Satellite Images

An optical earth observation satellite product was used in this study to provide spectral information for tableland area mapping. Sentinel-2, launched on 23 June 2015 by the European Space Agency (ESA), has an advanced spatial resolution of 10–60 m (depending on different bands) compared to the commonly used Landsat series (30 m). The high spatial resolution supports the selection of Sentinel-2 images in this study. We downloaded the Sentinel-2 images collected in June 2018 for tableland area mapping

because the image quality was good in June, with less cloud cover across the study area. Pre-processing of the seamless mosaic was then performed to generate a complete dataset covering the whole study area.

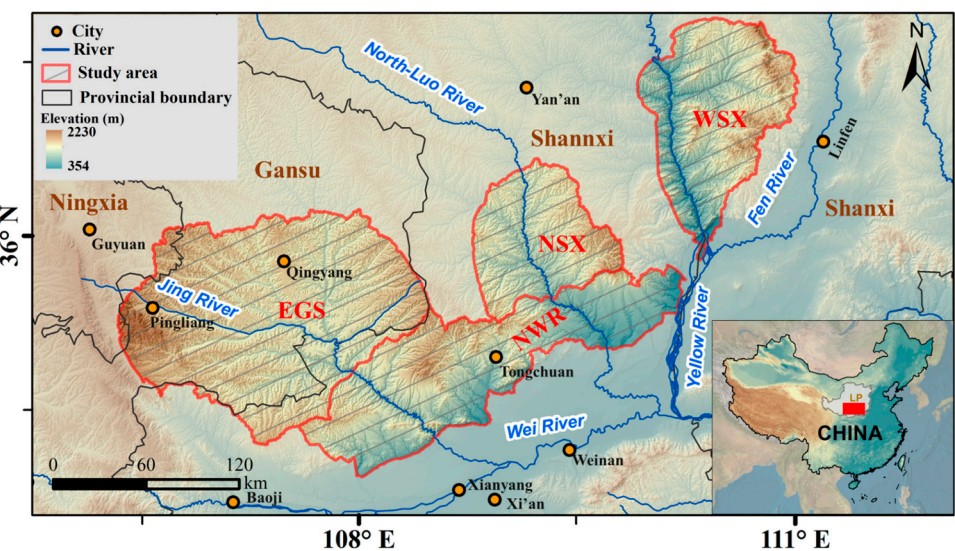

**Figure 2.** The distribution of study area and four subregions including eastern Gansu (EGS), northern Wei River (NWR), northern Shaanxi (NSX), and western Shanxi (WSX).

### 2.2.2. AW3D30 DEM

Digital elevation models (DEMs) are the basic input data in geomorphologic study. Previous studies have proven that topographic information derived from DEMs is critical for enhancing the performances of object-based extraction of landform elements [26,29]. AW3D30 DEM released by Japan Aerospace Exploration Agency (JAXA) was applied in this study, which was generated by Panchromatic Remote-sensing Instrument for Stereo Mapping (PRISM) onboard the Advanced Land Observing Satellite (ALOS), with an approximately 30 m horizontal resolution (basically 1 arc second). The latest version 3.2 released in March 2021 was used in this study. Although several public DEMs are available worldwide, such as SRTM DEM, ASTER GDEM, and TanDEM-X DEM, AW3D30 is superior to others with the best vertical accuracy proven by the existing literature [30–32].

### 2.2.3. Land-Use Data

In order to reveal the land-use change on the extracted loess tableland during the past decades, multi-temporal land-use data (the years of 2000, 2010, 2020) with 30 m spatial resolution, GlobeLand30, were used. This dataset includes ten land cover classes in total, namely cultivated land, forest, grassland, shrubland, wetland, water bodies, tundra, artificial surface, bare land, perennial snow, and ice [33]. The land cover classification is based on multiple remote sensing images, including TM5, ETM+, and OLI multispectral images of Landsat by NASA, HJ-1 (China Environment and Disaster Reduction Satellite), and GF-1 (China High-Resolution Satellite) multispectral image. Validation by random sampling comparison with field works indicates that the total accuracy is 83.50% (with a Kappa coefficient 0.78) for 2010, 85.72% (Kappa 0.82) for the year 2020, respectively.

### 3. Methods

The basic idea of the proposed tableland mapping method is based on the remarkable spectral and topographic feature difference between loess tableland areas and non-tableland areas (terraces and gullies). Generally, the tableland area has a gentle slope, while the area below the tableland boundary is characterized by a steep slope with intense soil loss. The mainstream object-based image analysis (OBIA) method combined with the machine learning algorithm was applied for distinguishing the tableland and non-tableland areas. In

addition to the traditional OBIA-based classification, the stream channels mainly originated from gully-affected areas were regarded as the terrain skeleton due to their clear implication of distinguishing gully and inter-gully objects. The workflow of the proposed approach includes four main steps: (1) data preparation and training dataset generation, (2) object-based extraction of loess tableland using the random forest classifier, (3) drainage network generation which matched with the gully-affected areas, and (4) initial error removal based on the spatial relationship between the drainage networks and segments (Figure 3).

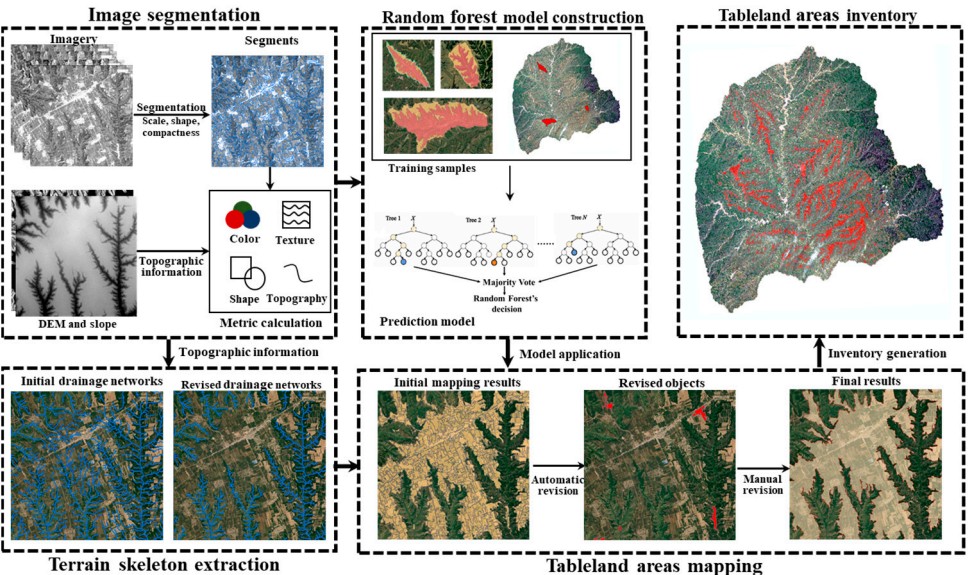

**Figure 3.** Overflow workflow of proposed method for loess tableland area mapping.

### 3.1. Data Preparation and Training Dataset Generation

As introduced in Section 2.1, the whole study area was divided into four subregions based on the landform similarity. Such division determines the application scope of prediction models so that the influence of spatial heterogeneity can be lowered. The training data selection is the critical step under the OBIA framework. In this study, 40 watersheds extracted by a 10 km$^2$ threshold value were selected. Generally, the selected training samples are in an even distribution while only occupying a small ratio of the study area. Although the large ratio of the training set may improve the overall accuracy, it is not recommended in practical use due to the highly time-consuming process of visual interpretation [32]. For each training sample, the tableland and non-tableland areas were distinguished manually by referring to remote sensing imagery. In addition to the training samples, each subregion owns two test samples determined by using a 100 km$^2$ threshold value (Figure 4).

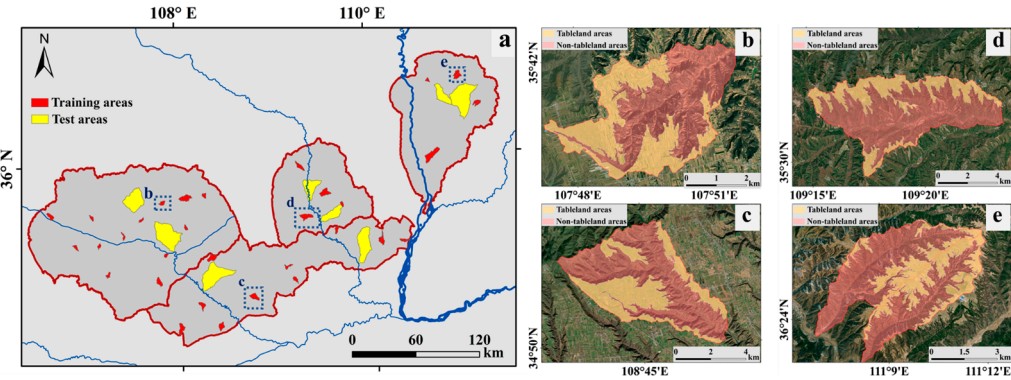

**Figure 4.** The distribution of training and test samples over the study area. (**a**) The selected training and test areas; (**b**–**e**) four enlarged areas in each subregion.

### 3.2. OBIA-Based RF Mapping on Tableland Areas

Object-based tableland area mapping was implemented in two sub-steps: image segmentation and classification of the segments. Image segmentation partitions the scene into a set of homogeneous objects. From these objects, it is possible to acquire more information such as shape and texture in addition to spectra, which is beneficial for improving the classification accuracy [34]. The eCognition Developer (v9.2) that integrated the multi-resolution segmentation algorithm was used for image segmentation. Scale is the most important image segmentation parameter, which directly controls the object size and influences the classification accuracy. Many studies have discussed the optimization strategy of scale parameters [29,35]; however, these strategies are not suitable for large-scale processing because the optimized parameter derived from training samples cannot ensure its feasibility within the whole study area due to strong spatial heterogeneity. Previous studies claimed that over-segmentation is more appropriate than under-segmentation because the over-segmented objects can be merged in later classification. In contrast, a small feature contained in a large object can no longer be detected [36]. In addition, as for the small training set ratio adopted in this study, a fine-scale segmentation is recommended according to previous studies [37]. This study adopted a unified scale parameter (120) after verifying that such scale parameter may bring over-segmentation phenomena based on the preliminary experiment conducted on the training areas. Meanwhile, shape and compactness were set to 0.5 and 0.3, respectively, referring to our previous studies [23,26].

The metric calculation was also conducted after image segmentation. Four kinds of features, including spectral, textural, geometric, and topographic information, were calculated within each object (Table 1). In addition to the commonly used spectral information, eight texture features derived from the Grey Level Co-occurrence Matrix (GLCM) were selected. Eight GLCM derivatives were calculated based on the red band in eCognition by adding the four directional values to obtain the rotation-invariant features. Geometric features are also widely used in the object-based landform mapping. We selected eight geometric features by referring to the previous study [23,38]. Slope value was used because of the huge difference between the flat tableland areas and steep gully-affected areas. The random forest method, a widely used ensemble classifier with high classification accuracy and processing efficiency, was further employed [39] to build the prediction model based on our previous evaluation [40]. We also noticed that some advanced algorithms are recommended for landform mappings, especially for regions with heterogeneous landscapes [41,42]. Although the spatial heterogeneity within the study area is also obvious, the zonal processing can ensure the high landform similarity within each subregion. Meanwhile, the task of this study to distinguish the tableland and non-tableland is relatively simple, rather than a multiclass mapping.

**Table 1.** Overview of features used for tableland area mapping.

| Feature Type | Feature Name |
| --- | --- |
| Spectral information | Mean band value (red, blue, green, and NIR) <br> Band rations (red/blue, blue/green) <br> Mean brightness <br> Maximum difference index |
| Textural information | GLCM Angular Second Moment <br> GLCM Contrast <br> GLCM Correlation <br> GLCM Dissimilarity <br> GLCM Entropy <br> GLCM Homogeneity <br> GLCM Mean <br> GLCM Standard Deviation |

**Table 1.** *Cont.*

| Feature Type | Feature Name |
|---|---|
| Geometric information | Area |
| | Length |
| | Length-width |
| | Roundness |
| | Asymmetry |
| | Compactness |
| | Rectangular Fit |
| | Shape index |
| Topographic information | Mean slope |

The training samples were generated by referring to the digitized tableland boundary. The created objects located in training areas were assigned to tableland or non-tableland labels based on the overlapping area of each type. Two key parameters of the random forest model, namely nTree (the number of trees that will grow) and mTry (the number of variables randomly sampled at each split), were set to 500 and 5, respectively [23]. The constructed prediction model was used to process the unlabeled objects, thereby generating the initial classification results.

*3.3. Terrain Skeleton Utilized for Error Correction*

In addition to the commonly used OBIA-based method, a topographic correction algorithm using drainage networks was applied. The flat terrains determine that the streams or rivers are unlikely to occur in tableland areas. In contrast, the stream headwaters are mainly located in the gully-affected area, which is featured by steep topography. Based on this natural law, the spatial relationship between the drainage networks and the segments can be applied for error correction. The key prerequisite of this revision processing is to generate drainage networks that match well with the gully-affected areas. Firstly, the drainage networks were generated based on DEMs with a small threshold value. The initial extracted drainage networks are spread over the tableland areas. Secondly, the tracking analysis was conducted from the source node along the flow direction. If the medium slope value within a $3 \times 3$ window is more than $10°$, the center pixel can be regarded as the inflection point between the tableland and gully-affected areas. Finally, the drainage networks were revised according to the detected stream headwaters (Figure 5).

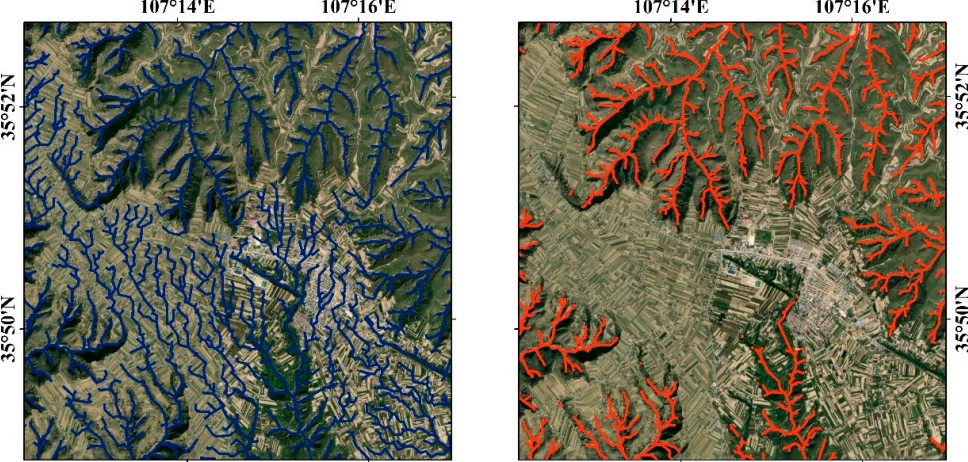

**Figure 5.** Comparison between the initial drainage networks (**left**) and the revised drainage networks (**right**) which are regarded as the terrain skeleton for modifying the mapping results.

After updating the drainage networks, two kinds of errors needed to be removed, including the omission errors in the tableland areas and commission errors in the non-tableland areas. The revision ruleset can be expressed as the following scenarios:

1.　If the non-tableland labeled object does not intersect with any drainage networks and the mean slope value is less than 5°, the object will be corrected to tableland.
2.　If the tableland labeled object intersects with any stream network and the mean slope value exceeds 10°, it will be corrected to non-tableland.

It should be noted that terrain skeleton-based correction is an auxiliary strategy after the OBIA-based classification. This strategy is promising to identify the mislabeled objects in some local areas, while it cannot be applied directly without the initial classification results for the whole region. Meanwhile, using slope values can reduce the uncertainty of terrain skeleton-based correction.

### 3.4. Validation and Final Product Generation

The performance of the proposed OBIA-based approach can be validated on the basis of the manually interpreted reference data. In this study, precision (Pr), recall (Re), F-score (F1), overall accuracy (OA), and Kappa coefficient (k) were adopted in the accuracy assessment. These metrics can be calculated as follows:

$$\text{Pr} = \frac{TP}{TP + FP} \tag{1}$$

$$\text{Re} = \frac{TP}{TP + FN} \tag{2}$$

$$\text{F1} = \frac{2 \times Pr \times Re}{Pr + Re} \tag{3}$$

$$\text{OA} = \frac{TP + TN}{TP + FP + TN + FN} \tag{4}$$

where *TP*, *TN*, *FP*, and *FN* denote the true positive, true negative, false positive, and false negative amount with respect to the reference, respectively.

The manual interpretation was conducted based on the automatic mapping results for generating the final inventory of tableland areas. Commission and omission errors were easy to remove by changing the classification results directly (Figure 6a,b). Special processing should be conducted fr = ir some under-segmented objects which contain both tableland areas and gully-affected areas (Figure 6c,d). In this case, the manual edition is needed to divide the large object into two different parts. Our team, familiar with loess landforms, assisted in the collection of the final data production.

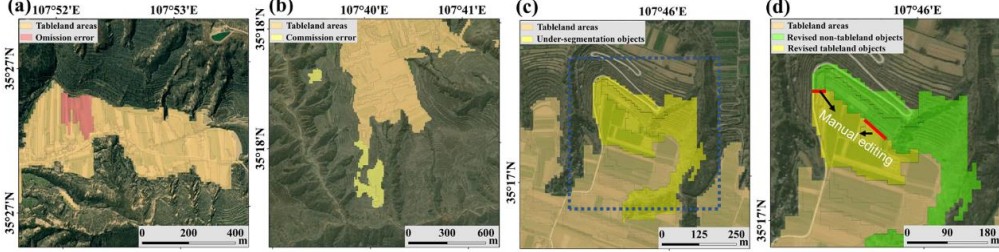

**Figure 6.** Manual editing for generating the inventory of loess tableland areas. (**a,b**) Commission and omission errors are removed by changing the type of segments. (**c,d**) Manual delineation sometimes is necessary if the object is under over-segmentation.

### 3.5. Detection of Erosion-Vulnerable Hotspots within Tableland Areas

In addition to providing a plateau-scale inventory of tableland distribution, this study also focuses on detecting the vulnerable areas for gully development. Although the tableland areas generally are featured by their broad areas, intense gully erosion may cause

a narrow shape in some local areas. If the gully developments continuously encroach the tableland without any manual interventions, the narrow area will likely be a breakthrough, leading to the split of original tableland areas. Such a phenomenon may damage the road connected to the nearby villages, with a dramatic increase in the transportation cost. The following steps detected the erosion-vulnerable areas. Firstly, a buffering area was calculated for a given gully head with a 500 m distance (Figure 7a). Secondly, the nearest gully head located in different catchments was regarded as the candidate point among all the selected points (Figure 7b). Thirdly, after generating the gully head pairs, manual interpretation was conducted to measure the narrowest width of the tableland areas threatened by the selected two gully heads (Figure 7c). All the detected risk points were classified into five levels based on the width. The classification criterion is mainly based on the expert knowledge and field investigation.

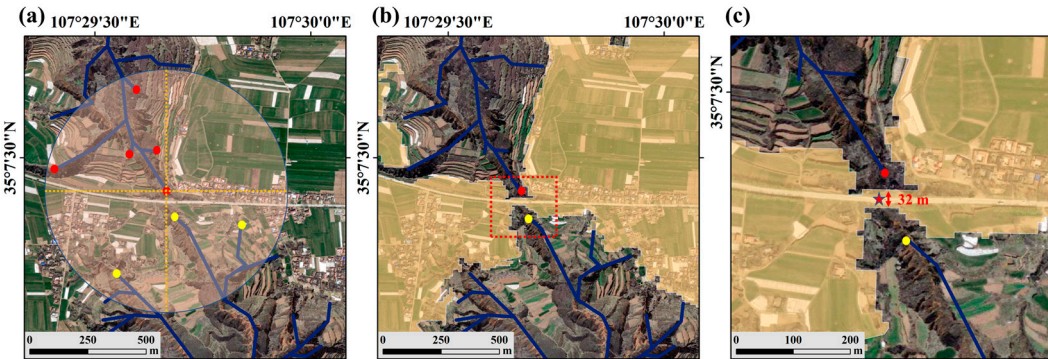

**Figure 7.** Steps for identifying the vulnerable areas for gully developments. (**a**) Potential gully heads were selected within a buffer zone. (**b**) The gully head pairs located in different catchments were detected. (**c**) The erosion-vulnerable area and its width was determined with manual interpretation.

## 4. Results

### 4.1. Accuracy Assessment of the Automatic Mapping Results

The accuracy of automatic mapping results was validated using the final manual-assisted tableland distribution maps. Comparison between mapping results using two strategies indicated that terrain skeleton revision improves the overall performances (Tables 2 and 3). This can be illustrated briefly by referring to the accuracy metrics in the region of eastern Gansu. This region's overall accuracy and kappa increased from 89.11% and 0.78 to 92.75% and 0.85, respectively. The initial user's accuracy (UA, also referred to as "commission error") of the tableland class was 81.89%, but it improved to 93.46% after revision. This demonstrated that the terrain skeleton information has successfully removed the overestimated tableland areas, which mainly are near the river valley. By contrast, the initial producer's accuracy (PA, also referred to as "omission error") decreased from 96.11% to 89.27%, indicating that the terrain skeleton also imported more omission errors. The specific characteristic of the northern Wei River is the complete tableland area with slight gully erosion. The initial OBIA-based mapping distinguishes tableland and non-tableland areas well with the overall accuracy and kappa of 94.55% and 0.85, respectively. However, the terrain skeleton revision fails to improve the accuracy further. The gully's extents and widths are very narrow in this region; hence, the influence of the limited DEMs' accuracy means the extracted drainage networks cannot match well with non-tableland areas.

**Table 2.** Accuracy assessment of the results without referring to terrain skeleton.

| Region | PA (%) | UA (%) | OA (%) | Kappa |
|---|---|---|---|---|
| West Shanxi | 67.47 | 58.88 | 90.36 | 0.57 |
| Northern Shannxi | 85.97 | 93.11 | 91.91 | 0.83 |
| Eastern Gansu | 81.89 | 96.11 | 89.11 | 0.78 |
| Northern Wei River | 97.04 | 95.83 | 94.55 | 0.85 |

**Table 3.** Accuracy assessment on the results by using topographic information.

| Region | PA (%) | UA (%) | OA (%) | Kappa |
|---|---|---|---|---|
| West Shanxi | 86.02 | 56.53 | 92.69 | 0.64 |
| Northern Shannxi | 93.88 | 87.58 | 93.36 | 0.85 |
| Eastern Gansu | 93.46 | 89.27 | 92.75 | 0.85 |
| Northern Wei River | 96.75 | 87.71 | 91.46 | 0.83 |

*4.2. Spatial Distribution of Tableland Areas across the Loess Plateau*

The final tableland area product of the Chinese Loess Plateau at 10 m resolution is available at zenodo repository [43]. Figure 8 shows the spatial distribution of tableland areas over the Chinese Loess Platea in 2018. The northern Wei River accounts for more than 48% (4570.46 km$^2$) of the total area (9507.31 km$^2$), followed by eastern Gansu (3587.93 km$^2$) and northern Shaanxi (996.01 km$^2$). In contrast, the remaining tableland areas in western Shanxi are only 315.40 km$^2$. In addition to the total area, we further quantified the tableland distribution by calculating the ratio between the tableland area and the total area in each hexagon with an area of 100 km$^2$. The spatial pattern verifies that the tableland areas are reserved well in the northern Wei River due to the slight gully erosion. The tableland areas occupied more than 50% of the total area with a maximum value of 89%. In contrast, the residual tableland areas in West Shanxi only account for less than 20% of the total area.

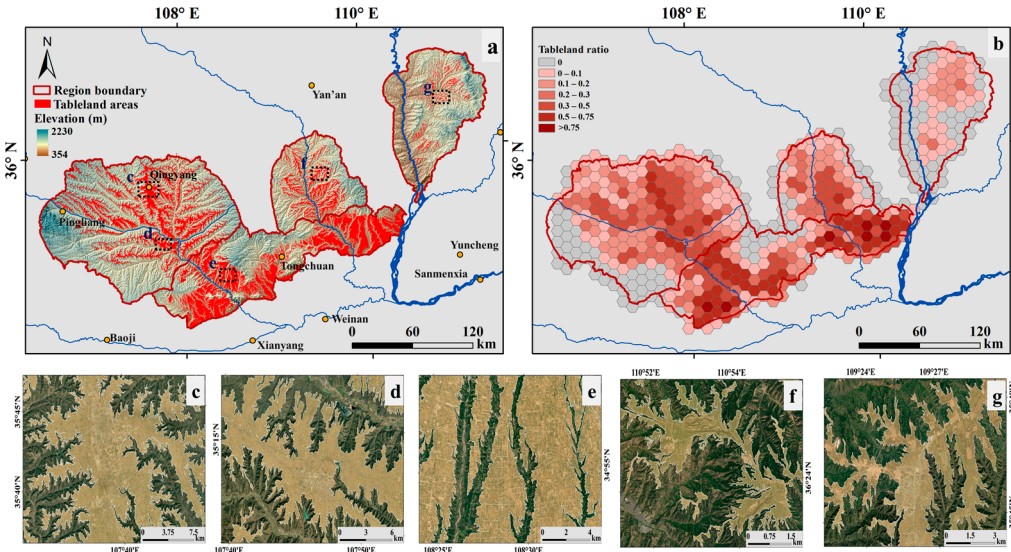

**Figure 8.** Distribution pattern of the extracted tableland areas. (**a**) Spatial distribution of tableland areas over the Loess Plateau. (**b**) Spatial variation of the tableland ratio. Zoom-in of five subregions are shown in (**c**) Qingyang, (**d**) Changwu, (**e**) Chunhua, (**f**) Luochuan, and (**g**) Daning.

Visually, our generated map realistically distinguishes well the tableland and non-tableland areas. As shown in Figure 8d, the gully density in the northern Wei River is very low, making the tableland areas well-preserved. In contrast, the gully-affected areas occupy a much higher percentage of the total area in eastern Gansu and northern Shaanxi. Although tableland areas suffer from serious gully erosion, the remaining flat landforms

still provide precious land recourses for agricultural production and human settlements. Some cities are located in the tableland areas, such as Qingyang, Changwu, and Luochuan (Figure 8b–e). Human activities in these regions may accelerate gully development, making the tableland areas risk shrinkage. A further potential situation is observed in western Shanxi, where the remaining tableland areas are limited due to the broken topography. As shown in Figure 8f, tableland areas cannot provide enough urban construction; hence, cities in this region are all located in the river valley.

We further investigated the topological features of the tableland areas by using the slope and elevation (Figure 9). The skewed distributions of the slope value derived from frequency histograms verified the flat of tableland areas. The mean slope of extracted tableland areas in eastern Gansu is only 3.57°, which is the lowest among the four subregions. Influenced by the fragmented tableland areas, western Shanxi's mean slope is relatively higher, with a value of 6.11°. The elevation values among tableland areas exhibit significant spatial heterogeneity among the four subregions. Generally, the tableland areas located in northern Shaanxi and western Shanxi follow unimodal distributions, which conform to the flat terrains of tableland landforms. In contrast, the elevation distributions within the eastern Gansu and northern Wei River own relatively wide ranges. Such two regions generally show a south-to-north increased pattern of elevation distributions.

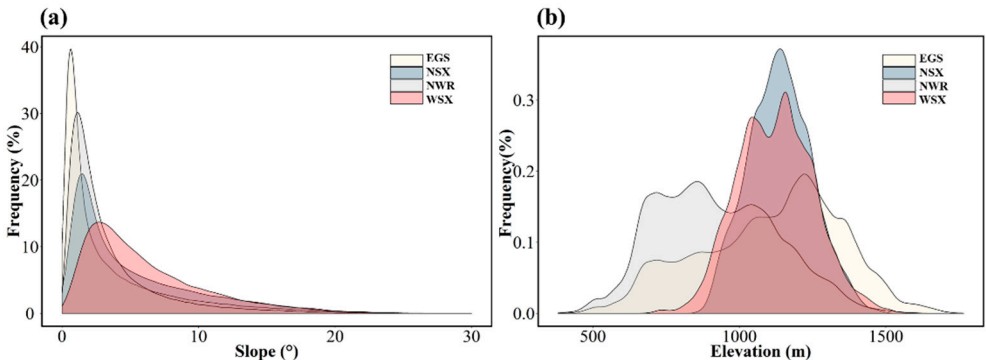

**Figure 9.** The frequency distribution of slope (**a**) and elevation (**b**) on the tableland areas.

*4.3. Land-Use Changes on Tableland Areas*

The area statistics of each land types are listed in the Table 4, and the spatiotemporal characteristics of land-use changes across tableland areas could be released after comparing the area change between the year 2020 and 2000 (Figure 10). Generally, the main land-use types on tableland areas were cultivated land, forest, grassland, and artificial surface, accounting for more than 99% of the total area in three different years. Among them, cultivated land has experienced a shrinkage trend since 2000, decreasing from 8466.82 km$^2$ to 7975.76 km$^2$. By contrast, forest, grassland, and artificial surface were increased by 57.53%, 14.33%, and 73.10%, respectively (Table 4). The increase in forest and grassland can be explained by the "Grain for Green" large-scale revegetation program operated since 1999. The increased amplitude in forest and grassland on the tableland area is not the whole story of such a massive vegetation plantation, because the revegetation programs were mainly implemented in the gully-affected areas. The expansion of the artificial surface was clearly represented in two selected examples, namely Qingyang and Luochuan. The rapid growth of the urban regions intensified the shortage of land resources in the tableland areas. As more urban architectures were built near the boundaries between tableland and gully-affected areas, the potential gully development (e.g., gully headcut, flank retreat) may threaten the human settlements.

**Table 4.** Statistics of land-use changes between 2000 and 2020 (km$^2$).

|  | 2000 | 2010 | 2020 |
| --- | --- | --- | --- |
| Cultivated land | 8466.82 | 8465.71 | 7975.76 |
| Forest | 577.74 | 512.63 | 910.1358 |
| Grassland | 264.60 | 311.74 | 302.54 |
| Artificial surface | 150.36 | 168.90 | 260.28 |
| Others | 47.48 | 48.02 | 58.29 |

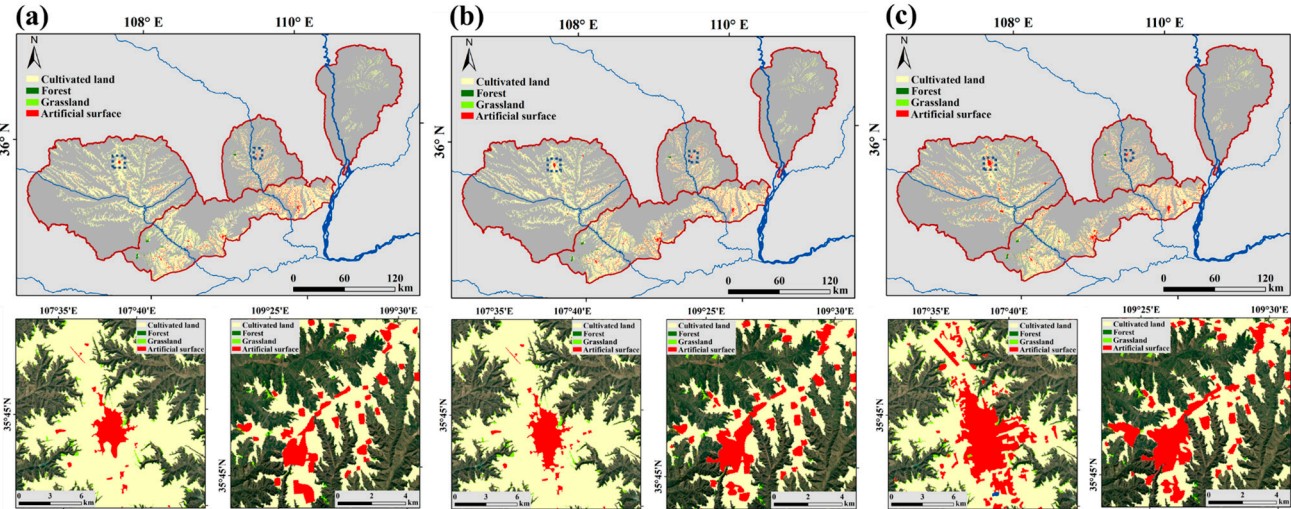

**Figure 10.** The land use on tableland areas in (**a**) 2000, (**b**) 2010, and (**c**) 2020. Two enlarged areas are Qingyang (**left**) and Luochuan (**right**), respectively.

*4.4. Distribution of Erosion-Vulnerable Hotspots*

After visually checking all the candidate risk points after automatic extraction, the final map was generated (Figure 11). A total of 455 risk points were detected, and nearly half of them (210) are distributed in eastern Gansu. The minimum width of tableland areas near the detected risk point was used for evaluating different risk levels. The spatial heterogeneity of the risk level is obvious in four different regions. Although western Shanxi only possesses 69 risk points, more than 60% of the points are classified into the highest level, with a width less than 20 m. This phenomenon suggests the high urgency of tableland protection in this region; otherwise, the residual tableland areas will become more fragmented and evolve into loess ridge landforms. Although the detected risk points in eastern Gansu are generally wider than those in western Shanxi, special attention and protective measures must be taken. This is because tableland areas in this region possess many living settlements. The split of a large tableland may greatly increase the transportation costs and even cause a geological disaster. Nevertheless, the protection task of tableland areas in the northern Wei River is not under a state of emergency. Among 60 detected risk points, only 12 cases need special treatment.

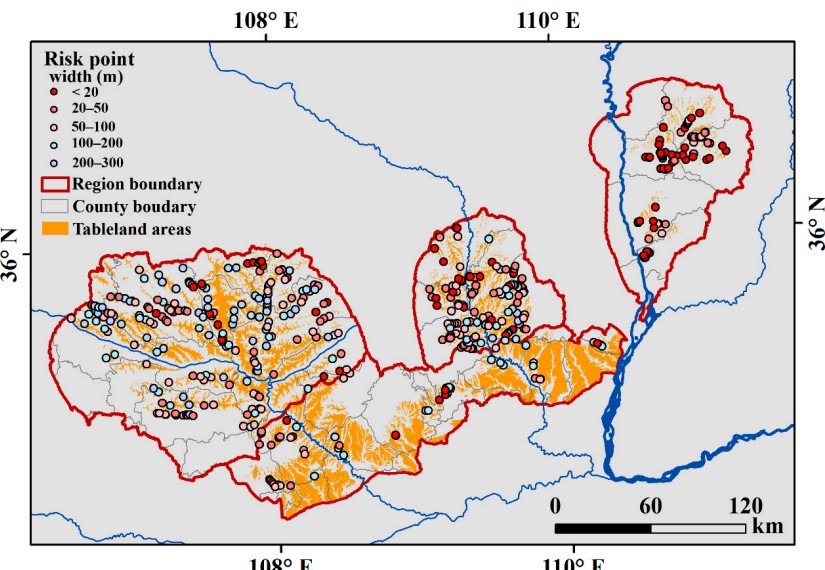

**Figure 11.** Distribution of extracted risk points over the study area.

## 5. Conclusions

This study is the first attempt to provide a trustable plateau-scale inventory of loess tableland areas in compliance with the SDG 15.3.1 goal based on EO data. Preliminary results indicate that the OBIA framework has helped the semi-automatic mapping of loess tablelands and the risk assessment. The implementation successfully demonstrated benefits, limitations, and needs for further developments to consolidate the approach. The approach has proven the large scalability, strong robustness, and reproducibility concerning the evidenced benefits. This enables more effective and efficient analysis if multiple time series and higher resolution EO data were introduced into future work. In particular, this total workflow can be executed every few years. The time series of results will consequently contribute to land dynamic monitoring service and land degradation assessment. This is a fundamental prerequisite to efficiently embed science into decision-making and avoid disseminating fake information in a post-truth world. Additionally, open access to our results is essential to support more open and reproducible research studies. Publishing the model output regularly at the pre-pixel level in the future might be a good complement to traditional national statistics since it provides spatial information as well, and more importantly, dynamic change by time series. We hope this dataset might be of help to large-scale studies in the related community and introduce the EO into a deeper understanding of the earth processes.

Currently, it is not possible to benefit from Sentinel-2 data for an earlier history period tableland mapping because the time series is not sufficiently long for this satellite series product. Landsat series can provide a good time series with relatively long historical information. However, to obtain reliable results, the 30 m resolution imagery is still coarse compared with the tableland shrinkage rate (Figure 12). We believe the mapping result can provide the potential users with a basement dataset for further analysis. The limitation of the EO data determines that the main contribution of this study is to provide the plateau-scale inventory at a benchmark year rather than long-term monitoring. The future update task could be performed by using multi-temporal UAV photogrammetry [44,45]. The collected high-resolution imageries and DEMs enable the effective monitoring of the erosion-vulnerable area. Considering the high cost and effort for large-scale data generation based on the UAVs, satellite images with very high spatial resolution such as WorldVeiw, QuickBird, GeoEye, and Gaofen serial might be the optimal choice for such kind of task. DEMs were also engaged in mapping tasks, but the data representing precious current terrain do not update very often. The update frequency mismatch between optical earth observation and terrain modeling might cause some information loss and mapping errors.

However, the proposed method in this paper only applies the DEM-derived stream network as the terrain skeleton information for error elimination. The DEM collection time and resolution do not significantly influence the final results.

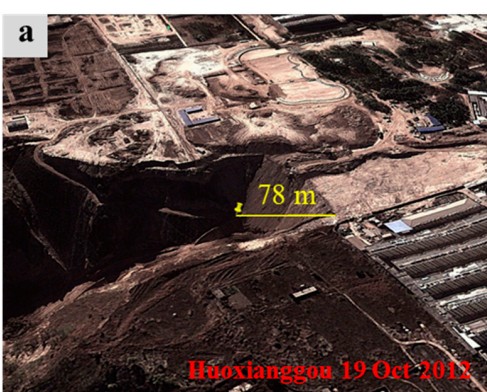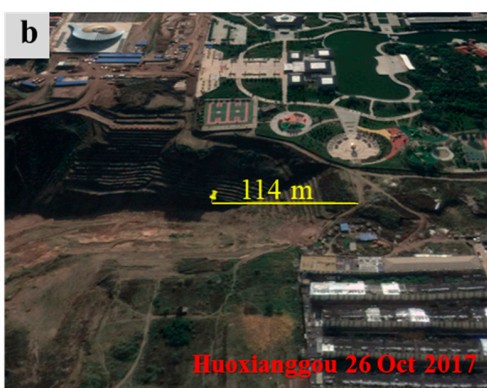

**Figure 12.** A typical region suffers from the tableland shrinkage. (**a**) Imagery captured on 19 October 2012; (**b**) imagery captured on 26 October 2017.

Over past decades, the land use and land cover of the Loess Plateau changed significantly in terms of both spatial patterns and area. This change was characterized by the "Grain for Green" project, leading to a decrease in farmland and an increase in vegetation cover and built-up areas. Among them, built-up areas mainly exist on the top of the loess tablelands, and the urbanization process is easily affected by the dynamic changes of tableland distribution. The Chinese government has already noticed this issue, and trustable monitoring is of great importance. Erosion-vulnerable hotspot detection will be useful for the land resource evaluation and risk assessment of human activity. Notably, the proposed identification of erosion-vulnerable hotspots of loess tablelands was still at a proof-of-concept stage. Its dynamic processes and potential application still need further exploration. Preventing tableland loss is an important issue, and land should be carefully managed to reduce gully erosion and land degradation while at the same time ensuring food security and sufficient provision of ecosystem services.

As for the actual practical solution to the erosion-vulnerable point of loess tablelands, vegetation and engineering on both the hillslope and the edge of the loess tableland may help. Specific protection measures, including returning farmland to forests and ecological restoration at the regional scale, are already in progress as a national policy. The governance for local areas should be carried out in the future, such as restraining the advance of ditch head engineering measures and preventing landslides at the edge of the loess tableland. Moreover, the relationship between land-use changes and erosion-vulnerable points should be deeply discovered and explored. Our initial implementation demonstrated that it is technically feasible to reveal the spatial pattern of tableland distribution that is essential to support efficient and effective land management on the Loess Plateau. The presented solution can help with actionable knowledge of soil and water conservation on the Loess Plateau and the global vision of land resource protection to achieve United Nations sustainable development goals.

To draw a very short conclusion of this study, we proposed the workflow for loess tableland mapping based on high-resolution imageries and DEMs with the strengthened capacities to use EO data. The spatial distribution of loess tablelands on the Loess Plateau of China was carried out, which is available for download at zenodo repository [43]. Moreover, the erosion-vulnerable hotspots were also evaluated and published. We hope this preliminary work can provide the essential information as a benchmark or basement for the ongoing project of tableland area protection. This study also demonstrates Earth Observations' potential capacities in supporting soil and water conservation practice at large scales. In addition to the contributions of this study, more efforts are still needed to overcome the existing limitations. Firstly, the accuracy of the OBIA-based classification could be

improved by using a more advanced machine learning algorithm and segmentation opti-mization strategy. Secondly, topographic correction strategy is not robust for all situations, especially when gullies are very narrow. Thirdly, our study only generates the plateau-scale inventory at a benchmark year rather than long-term monitoring. Continuous updates of the tableland areas will be of great significance.

**Supplementary Materials:** The following supporting information can be downloaded from https://www.mdpi.com/article/10.3390/rs14081946/s1. Figure S1: The regionalization map of soil erosion on the Loess Plateau which provides a coarse distribution of loess tableland (red line); Table S1: List of detected erosion-vulnerable areas.

**Author Contributions:** Conceptualization, K.L. and C.S.; methodology, K.L. and J.N.; software, C.F.; validation, Y.H., Z.W. and H.D.; formal analysis, K.L.; investigation, K.L., H.D. and J.N.; writing—original draft preparation, K.L. and J.N.; writing—review and editing, G.T. and C.S.; supervision, C.S.; project administration, C.S.; funding acquisition, K.L., G.T. and C.S. All authors have read and agreed to the published version of the manuscript.

**Funding:** This work was partly funded by the Strategic Priority Research Program of the Chinese Academy of Sciences (Grant No. XDA23100102), and National Natural Science Foundation of China (No. 41801321, 420013929, and 41930102).

**Data Availability Statement:** The tableland area product of the Chinese Loess Plateau at 10 m resolution is available at https://doi.org/10.5281/zenodo.5856924 (accessed on 16 January 2022). Sentinel-2 images were downloaded from: https://earthexplorer.usgs.gov (accessed on 30 June 2020). AW3D30 DEM is available at https://www.eorc.jaxa.jp/ALOS/en/aw3d30/data/index.htm (accessed on 30 June 2020).

**Acknowledgments:** The authors would like to present their appreciation to Jianing Yu and Bingyue Zhang for their contributions to validation.

**Conflicts of Interest:** The authors declare no conflict interest.

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
