# Peer review of "Large-Scale Detection of the Tableland Areas and Erosion-Vulnerable Hotspots on the Chinese Loess Plateau"

_remotesensing, doi:10.3390/rs14081946_

Round 1

Reviewer 1 Report

I would like to thank the authors for this interesting work. Please find my detailed comments below:

The abstract requires revision. The method and results need to be distinct. For example, in its present state, it is unclear as to what this “automatic mapping approach” entails and how classified image(s) from Sentinel-2 and DEM were obtained.

Introduction:

Lines 75-79: How is this automatic approach any different from what existing studies practice in tableland mapping? Specifically, how is it any different from existing classification approaches?

Line 84-89: It is unclear what heterogeneity the authors are referring to in this part. The term ‘spatial heterogeneity’ may refer to a widespread geomorphological processes or features. Please be specific as to how your study area is heterogeneous, as such that it creates spectral confusions in existing classification techniques and ML algorithms. Please mention any majority-minority bias in the classification process arising due to the geophysical heterogeneity of tableland.

Methods:

Why is Figure S1 necessary? What contribution does it make to this work, given Figure 2 already shows the study area?

Line 111: What is meant by “computation ability”? Spatial heterogeneity of what? Landforms or spectral attributes of the satellite image? Please specifically mention what aspects were considered for this categorization.

Line 119 and 124: What are the “some auxillary data”? Please clearly outline.

Line 120: Will it be “Globeland30” instead of “Globaland30”, please check all the terms? Please clearly mention the year of the raster used. Also, mention how each of these was used for erosion vulnerable hotspot identification.

Line 121: What are these “further” land-use change analyses?

Line 140: multitemporal – specifically mention the years

Section 2.2.3: The authors mentioned “some auxiliary data” in the previous sections, however, only the Globeland30 images were mentioned. Is the description of any dataset missing? If not, please revise and remove “some auxiliary data” and specifically mention Globeland30 data.

Lines 161-166: In Line 93, the authors mentioned that they adopted a “novel approach” . However, I could not find any novelty in this approach, this is the approach that are generally used for classification purposes. Also, given the authors has emphasized so much on the spatial heterogeneity that may complicate the classification process, it is surprising that the random forest classification was simply used without any feature selection process. Especially, when there are established literature regarding poor performances of RF in landforms with considerable heterogeneity. Please consult and discuss in your methods section the issue of RF and heterogeneity in light of the following articles:

https://www.mdpi.com/2072-4292/11/7/790

https://www.tandfonline.com/doi/pdf/10.1080/01431161.2014.903435?needAccess=true

https://ieeexplore.ieee.org/stamp/stamp.jsp?tp=&arnumber=5128907

Line 193: Justify these parameter selections. Either include a suitable reference or how these were decided.

Table 1: Please elaborately discuss Textural information and Geometric information. What datasets were employed?

The authors mentioned in Line 140 that multitemporal analyses were conducted. Couldn’t find a relevant methodological description.

Section 4.3 and 4.4: Proper methodological description required in the methods part. What was the process of vulnerability or risk assessment? How was the scoring method developed?

Figure 12: Please improve the caption to differentiate between the two images.

Author Response

Thank you for your positive feedback. Your detailed suggestions are beneficial for improving the quality of this manuscript.

Reviewer 2 Report

The results of the work are interesting and practically significant.

My main complaint is the lack of a (sub)section on the limitations and uncertainties of your study. Partially in the text, you wrote about it. But it would be desirable if you provided complete information about them at the end of the manuscript.

In addition:

Is the vertical scale in Figure 9b correct compared to Figure 9a?

Author Response

Thank you for your positive feedback. Your detailed suggestions are beneficial for improving the quality of this manuscript.

Q1. The results of the work are interesting and practically significant. My main complaint is the lack of a (sub)section on the limitations and uncertainties of your study. Partially in the text, you wrote about it. But it would be desirable if you provided complete information about them at the end of the manuscript.

Thank you for your suggestion. It is very essential to point out the limitation of the proposed method. We have already added more descriptions at the end of the manuscript as follows:

“Besides the contributions of this study, more efforts are still needed to overcome the existing limitations. Firstly, the accuracy of the OBIA-based classification could be improved by using a more advanced machine learning algorithm and segmentation optimization strategy. Secondly, the topographic correction strategy is not robust for all situations, especially when gullies are very narrow. Thirdly, our study only generates the plateau-scale inventory at a benchmark year rather than long-term monitoring. Continuous updates of the tableland areas will be of great significance.”

Q2. In addition: Is the vertical scale in Figure 9b correct compared to Figure 9a?

Thank you for your careful reading. We have checked this figure and confirmed that the vertical scales of both two figures are correct. We also noticed that the maximum value of the vertical figure 9b is much smaller than that in figure 9a. It is because the elevation values have much more unique values than the slope data.

Reviewer 3 Report

The paper is well structured and clear in its objectives and results. I only suggest to extend the bibliography by adding the following papers as well:

Lazzari M. 2020 - High-Resolution LiDAR-Derived DEMs in Hydrografic Network Extraction and Short-Time Landscape Changes. In: Gervasi O. et al. (eds) Computational Science and Its Applications - ICCSA 2020. Lecture Notes in Computer Science, vol 12250, pp 723-737. Springer, Cham. https://doi.org/10.1007/978-3-030-58802-1_52

Ionita, I., Fullen, M.A., ZgÅ‚obicki, W. et al. Gully erosion as a natural and human-induced hazard. Nat Hazards 79, 1–5 (2015). https://doi.org/10.1007/s11069-015-1935-z

Valentin C, Poesen J, Li Y (2005) Gully erosion: impacts, factors and control. Catena 63(2–3):132–153

Liu, K.; Ding, H.; Tang, G.; Na, J.; Huang, X.; Xue, Z.; Yang, X.; Li, F. Detection of Catchment-Scale Gully-Affected Areas Using Unmanned Aerial Vehicle (UAV) on the Chinese Loess Plateau. ISPRS Int. J. Geo-Inf. 2016, 5, 238. https://doi.org/10.3390/ijgi5120238

As regards, however, the other points, I can say the work does not have major criticalities. Some things to improve could be the following:

1. to clarify the concept and applicability of Spatial heterogeneity "

2. better specify how the multitemporal analysis was conducted

Author Response

(The authors gave the same response as above.)

Round 2

Reviewer 1 Report

I would like to thank the authors for the revised manuscript. It is substantially better now. I have just one suggestion for the authors, which they may or may not consider incorporating.

The answer to my comments, Response 2 and 3, regarding the novelty of the proposed approach and spatial heterogeneity in the study could be incorporated into the manuscript. The way these two aspects have been articulated in the responses reads much better than the way it has been written in the manuscript (even in the revised version). I believe such clear differentiation and explanation will facilitate the understanding of the readers and make the research more impactful. 

Author Response

Q1. I would like to thank the authors for the revised manuscript. It is substantially better now. I have just one suggestion for the authors, which they may or may not consider incorporating.

The answer to my comments, Response 2 and 3, regarding the novelty of the proposed approach and spatial heterogeneity in the study could be incorporated into the manuscript. The way these two aspects have been articulated in the responses reads much better than the way it has been written in the manuscript (even in the revised version). I believe such clear differentiation and explanation will facilitate the understanding of the readers and make the research more impactful.

A1: We sincerely thank the reviewer for the positive feedback on the revised manuscript. Your suggestions have been adopted. More details of the novelty of the proposed approach and spatial heterogeneity have been incorporated into the manuscript. We hope that the main contribution of this study could be clearly conveyed to readers.

“A novel approach was designed for large-scale tableland areas mapping using a two-step classification. The first step of the proposed method is similar to the existing remote-sensing method in which object-based image analysis and random forest algorithms were used. After achieving the initial classification results, we further used the drainage networks derived from the DEMs to correct the classification results. The topographic correction strategy is based on the truth that rivers or streams are unlikely to occur in tableland areas due to the flat terrains.” (Line 98-105) 

“It should be noted that the geomorphological features of the tableland areas within the study area exhibit strong spatial heterogeneity due to the different gully development stages. For example, the gully density in the northern part of the study area is much higher than the southern areas. In this study, we divided the whole study area into four subregions based on the landform similarity, including eastern Gansu (EGS), northern Wei River (NWR), northern Shaanxi (NSX), and western Shanxi (WSX). The predicted model was constructed and applied within four subregions independently so that uncertainties caused by the geomorphic difference can be reduced largely.” (Line 128-137)